# Effects of long working hours on metabolic dysfunction-associated steatotic liver disease, with and without increased alcohol intake, in healthy workers: A 10-year cohort study

Yesung Lee[1], Woncheol Lee[2]*

1 Department of Occupational and Environmental Medicine, Medical Support Division, Pyeongchang County Public Health Clinic, Gangwon-do, Korea, 2 Department of Occupational and Environmental Medicine, Kangbuk Samsung Hospital, School of Medicine, Sungkyunkwan University, Seoul, Korea

* doctor.oem@gmail.com

## Abstract

Long working hours are increasingly recognized as a significant occupational hazard linked to various adverse health outcomes. This study aimed to assess how prolonged working hours relate to the subsequent development of metabolic dysfunction-associated steatotic liver disease (MASLD) and metabolic and alcohol-associated liver disease (MetALD), based on a newly proposed classification system. We analyzed data from 117,354 Korean employees who underwent at least two abdominal ultrasound examinations between 2012 and 2023. Long working hours referred to a weekly workload of at least 60 hours. The primary outcome was the incidence of steatotic liver disease, categorized as MASLD or MetALD. A total of 28,361 new cases were identified over 627,094 person-years of follow-up. Cox proportional hazards models were used to estimate adjusted hazard ratios (HRs) and 95% confidence intervals (CIs). Participants working ≥60 hours per week had a higher risk of developing MASLD (adjusted HR 1.17, 95% CI 1.03–1.32) and MASLD+MetALD (adjusted HR 1.18, 95% CI 1.05–1.33) compared with those working 35–59 hours per week. Subgroup analyses indicated that the association was especially pronounced among individuals aged ≤47 years and in men. These results suggest that prolonged working hours contribute to an increased risk of MASLD and MASLD+MetALD, particularly in younger and male populations, emphasizing the importance of occupational health interventions.

## Introduction

Long working hours are a well-established occupational risk factor associated with adverse health outcomes, and there is a global movement toward reducing working hours to protect workers' health and well-being [1–3]. However, in 2023, the South

which permits unrestricted use, distribution, and reproduction in any medium, provided the original author and source are credited.

**Data availability statement:** The data contain potentially identifying and sensitive health information from the Kangbuk Samsung Health Study and cannot be publicly shared under the conditions of the protocol approved by the Institutional Review Board of Kangbuk Samsung Hospital (IRB No. KBSMC2025-04-003). In compliance with institutional regulations, the dataset cannot be downloaded to personal computers; access is strictly limited to a secure virtual desktop infrastructure (VDI) environment provided by the institution. All researchers must obtain prior approval of their study protocol from the Institutional Review Board, and can only access the data permitted by the Cohort Research Center. Therefore, access to the data requires the approval of both the Institutional Review Board of Kangbuk Samsung Hospital and the Cohort Research Center, and de-identified data will be made available upon reasonable request through the designated institutional body. De-identified data will be made available upon reasonable request. Data inquiries can be directed to Ms. Suhyeon Moon (Academic Promotion Team, Kangbuk Samsung Hospital; email: suhyeon71.moon@samsung.com.).

**Funding:** This work was supported by the National Research Foundation of Korea (NRF) grant funded by the Korean government (MSIT) (RS-2023-00274176). The funders had no role in study design, data collection and analysis, decision to publish, or preparation of the manuscript.

**Competing interests:** The authors have declared that no competing interests exist.

Korean government announced a plan to reform the working-hour system. This reform plan allowed a maximum of 69 working hours per week [4]. However, this plan has faced strong criticism for running counter to global trends and has been suspended. Although South Korea has made efforts to reduce working hours over time [5], OECD data show that the national average still reached 1,872 hours in 2023, which is roughly 130 hours above the OECD average [6]. Consequently, health risks related to prolonged working hours continue to be a concern. Extended work hours have been implicated in increased susceptibility to cerebro-cardiovascular disease [7], which is formally designated as a work-related health outcome in South Korea [4]. Furthermore, recent studies have reported that long working hours are associated with an increased risk of non-alcoholic fatty liver disease (NAFLD) [8–10].

In 2023, an international multisociety Delphi consensus formally established the new nomenclature metabolic dysfunction–associated steatotic liver disease (MASLD), replacing the former NAFLD classification to better reflect the cardiometabolic components of steatotic liver disease [11,12]. MASLD is defined as the presence of hepatic steatosis together with at least one cardiometabolic risk factor in the absence of significant alcohol intake. In contrast, *metabolic and alcohol-associated liver disease* (MetALD) refers to cases where metabolic dysfunction coexists with moderate alcohol consumption. Although MASLD and MetALD share overlapping metabolic pathways, alcohol may act additively or synergistically to accelerate hepatic injury, underscoring the importance of differentiating these entities. This new classification system has broadened our understanding of the disease etiology and pathophysiology. MASLD has also emerged as a major global health concern, affecting nearly 38% of adults worldwide and projected to exceed 55% by 2040. Beyond liver-related outcomes such as cirrhosis and hepatocellular carcinoma, MASLD is strongly linked to cardiometabolic diseases, with cardiovascular disease being the leading cause of death in affected individuals [13]. In line with the new classification system, a recent cross-sectional study found that long working hours were significantly associated with MASLD among Korean male workers [14].

Although previous research has considered the potential link between prolonged working hours and NAFLD, prospective investigations targeting MASLD and MetALD are scarce. Therefore, this study aimed to clarify the temporal association between extended work hours and incident steatotic liver disease—including both MASLD and MASLD+MetALD—using a decade of cohort data.

## Materials and methods

### Study population

The Kangbuk Samsung Health Study is a cohort study of Koreans aged at least 18 years who attended a comprehensive health screening examination on an annual or biannual basis at the Kangbuk Samsung Hospital Total Healthcare Center in Seoul and Suwon, South Korea [15]. Most participants in the examination were employees of various companies, local government organizations, and their spouses. The Occupational Safety and Health Act of South Korea mandates free annual and biannual

health examinations for all employees. The other participants voluntarily underwent medical checkups at a healthcare center.

The Kangbuk Samsung Health Study is an ongoing cohort with continuous data collection. For this analysis, data up to December 31, 2023, were available, and Institutional Review Board (IRB) approval was obtained in 2025 after data linkage was completed. This study included 466,988 participants who underwent health examinations, including abdominal ultrasonography, between January 1, 2012, and December 31, 2022, and had undergone at least one other health check-up before December 31, 2023. First, we excluded 308,872 participants meeting any of the following criteria at baseline (Fig 1): missing data on ultrasonography or average working hours per week; working less than 35 hours per week; steatotic liver on ultrasonography; medication use for hepatitis or findings of liver cirrhosis on ultrasonography; positive serologic markers for hepatitis B or C; history of malignancy; missing data on alcohol intake; use of steatogenic medications, such as amiodarone, tamoxifen, methotrexate, valproate, or corticosteroids, within the past year; age > 65 years or < 19 years; and undergoing ultrasonography fewer than two times during the follow-up period. Second, among 158,116 workers who were potential participants, we further excluded 40,762 participants with a censoring period of less than 1

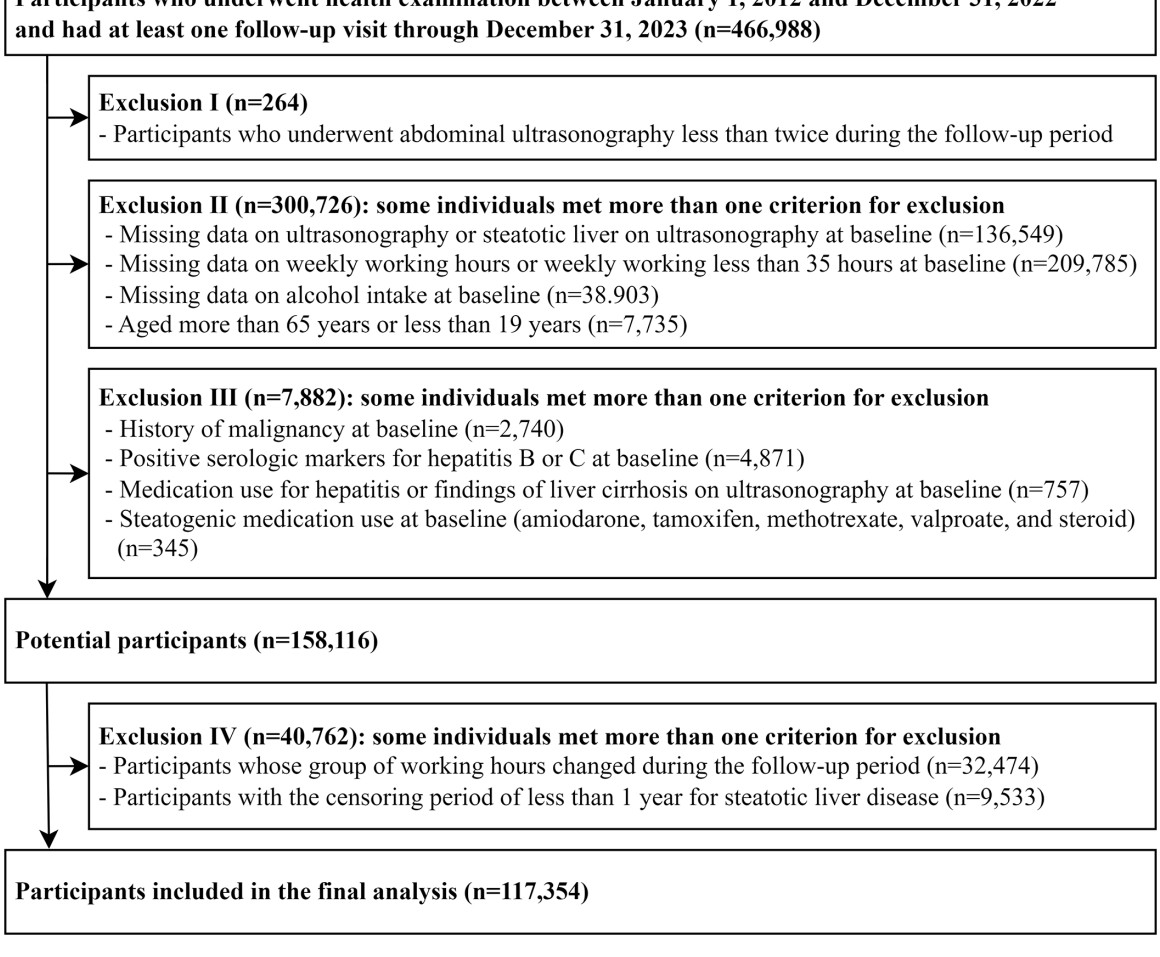

**Fig 1. Flow chart of study population selection with exclusion criteria.** Participants were excluded sequentially according to predefined criteria. Exclusions occurred throughout the study period (2012–2022) without temporal clustering, as further detailed in Supplementary Table S1, which presents the distribution of exclusions by category across 3-year baseline periods.

year for steatotic liver disease and participants whose group of working hours changed during the follow-up period to minimize the impact of fluctuations in their weekly working hours. Ultimately, 117,354 participants were eligible at baseline.

Our study was approved by the IRB of Kangbuk Samsung Hospital, which waived the requirement for informed consent because we accessed only de-identified data routinely collected during health screening examinations (IRB No. KBSMC2025-04-003). All methods were performed in accordance with the relevant guidelines and regulations. The dataset was accessed for research purposes on 26 April 2025. The dataset was fully anonymized before the authors accessed it. Therefore, the authors did not have access to any personally identifiable information during or after data collection.

## Measurements

All examinations were conducted at the Kangbuk Samsung Hospital Total Healthcare Screening Centers in Seoul and Suwon. At each visit, data on demographic and working characteristics, smoking status, alcohol consumption, regular exercise, education level, monthly household income, marital status, medical history, and medication use were collected using standardized self-administered questionnaires [15]. Smoking status was categorized as non-current and current smokers. Alcohol consumption was categorized as nondrinking, light drinking (<30 g/day for men and <20 g/day for women), moderate drinking (30–60 g/day for men and 20–50 g/day for women), and heavy drinking (>60 g/day for men and >50 g/day for women). Regular exercise was defined as engaging in vigorous physical activity during leisure time, at least three times per week. Educational level was categorized as less than college, college graduate, or higher. Monthly household income was categorized as <6 million Korean Won (KRW) per month or ≥6 million KRW per month. Marital status was categorized as married or unmarried. On the day of the health examination, a trained nurse verified the questionnaire for blanks, and a trained doctor double-checked the questionnaire for mistakes or blanks while conducting a face-to-face interview with the examinee at the final step of the health examination.

Working hours were assessed using the following question: "How many hours did you work in a week on average in your job for the past year, including overtime?" According to the Labor Standards Act of Korea, adult workers are only allowed to work for a maximum of 40 h/week, excluding breaks, and an additional 12 h/week is allowed with the approval of the employee. This does not apply to workplaces with fewer than five employees. Workers in exceptional industries, such as transportation and healthcare services, are permitted to work for more than 52 h/week. If a worker develops cerebro-cardiovascular disease after working more than 60 h/week for 12 weeks, it is classified as an occupational disease by public notice of the Department of the Ministry of Employment and Labor of Korea. We excluded part-time workers to minimize deviations in the evaluation of the health effects of long working hours. In accordance with the 2021 Labor Statistics of the Korea Labor Institute, a public institution under the Prime Minister, individuals who worked less than 35 h/week were defined as part-time workers. Based on the above-mentioned standards and the 90th percentile of weekly working hours in the study population (60 h/week), weekly working hours were categorized as 35–59 (reference group) and ≥60 h/week (long working hours group). This binary classification was chosen to highlight the health impact of long working hours in survival analysis, where a two-group comparison was considered more informative than multiple finer categories. The shift work schedule was categorized as daytime work (work performed mostly between 6 AM and 6 PM) or shift work (work performed during other hours).

Trained nurses measured blood pressure, weight, height, and waist circumference [15]. Obesity was defined as a body mass index (BMI) of ≥23 kg/m$^2$. Hypertension was defined as a systolic blood pressure of ≥140 mmHg, a diastolic blood pressure of ≥90 mmHg, a self-reported history of hypertension, or current use of antihypertensive medications. Diabetes mellitus (DM) was defined as a fasting serum glucose level of ≥126 mg/dL, a hemoglobin A1c (HbA1c) level of ≥6.5%, a self-reported history of DM, or current use of antidiabetic medications. Fasting blood parameters included glucose, total cholesterol, low-density lipoprotein cholesterol, triglycerides, and high-density lipoprotein cholesterol.

Abdominal ultrasonography was performed using a LOGIQ Q700 MR 3.5-MHz transducer (GE, Milwaukee, WI, USA) by experienced radiologists who were blinded to the study aims. Images were obtained in a standard fashion with the

patients in the supine position with their right arm raised above their heads [16]. The ultrasonographic diagnosis of steatotic liver disease was defined as the presence of a diffuse increase in fine echoes in the liver parenchyma compared to those in the kidney or spleen parenchyma [17]. The inter-observer and intra-observer reliabilities in the diagnosis of steatotic liver disease were very high (kappa statistics of 0.74 and 0.94, respectively) [18].

MASLD was defined as the presence of steatotic liver disease with one or more of the following criteria: (1) overweight or obesity by the Asia-Pacific criteria (BMI ≥ 23 kg/m²) or high waist circumference (>94 cm for men and >80 cm for women); (2) prediabetes or type 2 DM (fasting serum glucose levels of ≥100 mg/dL, HbA1c levels of ≥5.7%, history of type 2 DM, or treatment for type 2 DM); (3) blood pressure of ≥130/85 mmHg, history of hypertension, or treatment for hypertension; (4) plasma triglyceride levels of ≥150 mg/dL or specific drug treatment; (5) plasma high-density lipoprotein cholesterol levels of <40 mg/dL for men and <50 mg/dL for women or specific drug treatment. As participants with other identifiable causes of steatotic liver disease were excluded at baseline according to the exclusion criteria, incident steatotic liver disease cases in this study were classified as MASLD (with no alcohol or light alcohol intake), or MetALD (metabolic and alcohol-associated liver disease).

## Statistical analysis

Baseline characteristics of the study participants are presented according to the groups of weekly working hours. Descriptive statistics were used to summarize the characteristics of the participants grouped according to working hours. The primary endpoint was development of steatotic liver disease. Based on baseline data, we divided participants who developed steatotic liver disease into two subgroups: MASLD and MASLD+MetALD. Participants were followed up from baseline to the endpoint visit or to the last available visit until December 31, 2023, whichever came first. The incidence density was calculated as the number of incident cases divided by person-years of follow-up.

Hazard ratios (HRs) and 95% confidence intervals (CIs) for incident steatotic liver disease were estimated using Cox proportional hazards regression analyses. Initially, we adjusted for age and sex in the crude model. Model 1 was adjusted for smoking status and regular exercise. Model 2 was further adjusted for education level, marital status, household income, and shift work schedule to explore the social mechanisms underlying the observed associations between working hours and the risk of steatotic liver disease. We assessed the proportional hazards assumption using Schoenfeld residuals using the *estat phtest* command in Stata, and no violation of this assumption was detected.

To explore whether associations between long working hours and the risk of steatotic liver disease differed according to clinically relevant factors, subgroup analyses were performed by sex (men vs. women) and age groups (≤47 years vs. ≥48 years), based on the 90th percentile of age in the study population. To account for multiple testing across these subgroup comparisons, we applied Bonferroni correction, and both unadjusted and corrected p-values were reported. Although the number of participants working ≥60 hours per week was relatively small (n = 2,243), the overall large cohort size (>117,000 participants) provided sufficient statistical power to detect meaningful associations.

As a sensitivity analysis, we conducted a time-dependent Cox regression without excluding participants whose weekly working hours group changed during follow-up. In this model, weekly working hours were treated as a time-varying covariate, and participants with missing data on working hours at any visit were excluded. A total of 147,739 participants were included in this analysis, which was based on an alternative sample not shown in Fig 1. In the main analysis, participants whose working hours changed during follow-up were excluded to minimize exposure misclassification, and working hours were categorized into two groups (35–59 and ≥60 hours/week). In contrast, the sensitivity analysis used a three-category working hours variable (≤40, 41–52, ≥53 hours/week), allowing for time-varying exposure classification based on repeated measurements.

## Results

In Table 1, we analyzed baseline characteristics of 117,354 participants with 35–59 weekly working hours (reference group; mean [standard deviation] age, 38.0 [8.6] years) and those of 2,243 participants with ≥60 weekly working hours (long working hours group; mean [standard deviation] age, 34.4 [7.0] years). Among participants with long working hours,

**Table 1. Baseline characteristics of study participants by weekly working hours.**

| Characteristics | Weekly working hours | | P-value |
|---|---|---|---|
| | 35–59 | ≥60 | |
| Number | 115,111 | 2,243 | |
| Incident cases of MASLD | 25,374 (22.0) | 327 (14.6) | <0.001 |
| Incident cases of (MASLD+MetALD) | 27,982 (24.3) | 379 (16.9) | <0.001 |
| Age (years)* | 35.3 (7.9) | 37.7 (9.8) | <0.001 |
| Age group | | | |
| ≤47 years | 104,670 (90.9) | 1,844 (82.2) | <0.001 |
| ≥48 years | 10,441 (9.1) | 399 (17.8) | <0.001 |
| Men | 62,928 (54.7) | 1,382 (61.6) | <0.001 |
| Alcohol intake | | | |
| Non-drinking | 14,364 (12.5) | 258 (11.5) | <0.001 |
| Light drinking | 89,433 (77.7) | 1,654 (73.7) | <0.001 |
| Moderate drinking | 9,733 (8.5) | 271 (12.1) | <0.001 |
| Heavy drinking | 1,581 (1.4) | 60 (2.7) | <0.001 |
| Current smoker | 18,882 (16.4) | 621 (27.7) | <0.001 |
| Regular exercise[a] | 15,500 (13.5) | 304 (13.6) | 0.779 |
| High education level[b] | 97,473 (84.7) | 1,739 (77.5) | <0.001 |
| Marital status – married | 75,148 (65.3) | 1490 (66.4) | 0.097 |
| High household income[c] | 36,669 (31.9) | 817 (36.4) | <0.001 |
| Systolic BP (mmHg)* | 106.4 (11.8) | 107.3 (12.1) | <0.001 |
| Fasting glucose (mg/dL)* | 92.3 (10.3) | 93.3 (13.6) | <0.001 |
| Total cholesterol (mg/dL)* | 187.9 (31.8) | 190.5 (32.5) | <0.001 |
| BMI (kg/m²)* | 22.3 (2.8) | 22.7 (2.8) | <0.001 |
| Daytime worker[d] | 100,816 (87.6) | 1,680 (74.9) | <0.001 |

Data are expressed as * mean (standard deviation) or number (%).

[a] ≥ 3 times/week

[b] ≥ College graduate

[c] ≥ 6 million KRW per month

[d] Participants who answered "I work mostly during the day (between 6 AM and 6 PM)"

MASLD, metabolic dysfunction-associated steatotic liver disease, MetALD, metabolic and alcohol-associated liver disease; BP, blood pressure; BMI, body mass index; KRW, Korean Republic Won

there were higher proportions of men, moderate drinkers, heavy drinkers, current smokers, and shift workers. Participants with long working hours showed slightly higher levels of systolic blood pressure, fasting glucose, total cholesterol, and body mass index (BMI).

Table 2 presents the relationship between prolonged working hours and the incidence of steatotic liver disease. Of 117,354 participants in the reference group, 28,361 incident cases of steatotic liver disease (incidence density, 4.52/100 person-years) were identified over 627,094 person-years of follow-up (median follow-up, 5.3 years). Of the 28,361 incident cases, 25,701 were of MASLD. Across all analytical models, individuals working prolonged hours exhibited a higher incidence of steatotic liver disease—including both MASLD and MetALD—compared to those in the 35–59 hours per week reference category. In Model 2, which included further adjustment for additional sociodemographic variables, the adjusted HRs for MASLD and MASLD+MetALD among those working ≥60 hours per week, compared to 35–59 hours, were 1.17 (95% CI 1.03–1.32) and 1.18 (95% CI 1.05–1.33), respectively.

**Table 2. Development of steatotic liver disease according to weekly working hours.**

| Weekly working hours | Person-years (PY) | Incident cases | Incidence density (per 100 PY) (95% CI) | Age- and sex-adjusted HR (95% CI) | Multivariable-adjusted HR (95% CI)[a] | |
|---|---|---|---|---|---|---|
| | | | | | Model 1* | Model 2** |
| *MASLD* | | | | | | |
| 35–59 | 619,159 | 25,374 | 4.10 (4.05–4.15) | 1.00 (reference) | 1.00 (reference) | 1.00 (reference) |
| ≥60 | 7,935 | 327 | 4.12 (3.70–4.59) | 1.15 (1.03–1.28) | 1.13 (1.01–1.26) | 1.17 (1.03–1.32) |
| *MASLD+MetALD* | | | | | | |
| 35–59 | 619,159 | 27,982 | 4.52 (4.47–4.57) | 1.00 (reference) | 1.00 (reference) | 1.00 (reference) |
| ≥60 | 7,935 | 379 | 4.78 (4.32–5.28) | 1.19 (1.08–1.32) | 1.17 (1.05–1.30) | 1.18 (1.05–1.33) |

[a]Estimated from Cox proportional hazard models.

* Model 1 was adjusted for age, sex, smoking status, and regular exercise.

** Model 2: model 1 plus adjustment for education level, marital status, household income, and shift work schedule.

CI, confidence interval; HR, hazard ratio; MASLD, metabolic dysfunction-associated steatotic liver disease; MetALD, metabolic and alcohol-associated liver disease.

In subgroup analyses (Table 3), working ≥60 hours per week was significantly associated with an increased risk of MASLD among participants aged ≤47 years (HR 1.21, 95% CI 1.06–1.39). A similar pattern was observed for MASLD+Met-ALD in the same age group (HR 1.23, 95% CI 1.08–1.39). Among male participants, long working hours were significantly associated with MASLD+MetALD (HR 1.17, 95% CI 1.03–1.34), but not with MASLD alone (HR 1.15, 95% CI 0.99–1.32). After applying Bonferroni correction for multiple subgroup comparisons, the association remained statistically significant only in participants aged ≤47 years, whereas the associations by sex did not remain significance (Supplementary Table S2).

In the time-dependent Cox regression analysis including participants with changes in working hours over time, the association between longer working hours and MASLD remained consistent. Compared with those working ≤40 hours per week, the multivariable-adjusted hazard ratios (95% CIs) for MASLD were 1.077 (1.050–1.106) for 41–52 hours and 1.378 (1.322–1.435) for ≥53 hours (Table 4).

**Table 3. Hazard ratios[a] (95% CI) for steatotic liver disease by weekly working hours in clinically relevant subgroups.**

| Subgroup | Weekly working hours | |
|---|---|---|
| | 35–59 | ≥60 |
| *MASLD* | | |
| Women | 1.00 (reference) | 1.06 (0.81–1.37) |
| Men | 1.00 (reference) | 1.15 (0.99–1.32) |
| Aged ≤47 years | 1.00 (reference) | 1.21 (1.06–1.39) |
| Aged ≥48 years | 1.00 (reference) | 1.05 (0.75–1.46) |
| *MASLD+MetALD* | | |
| Women | 1.00 (reference) | 1.05 (0.81–1.35) |
| Men | 1.00 (reference) | 1.17 (1.03–1.34) |
| Aged ≤47 years | 1.00 (reference) | 1.23 (1.08–1.39) |
| Aged ≥48 years | 1.00 (reference) | 1.08 (0.79–1.46) |

[a]Estimated from Cox proportional hazard models adjusted for age, sex, smoking status, regular exercise, education level, marital status, household income, and shift work schedule.

CI, confidence interval; MASLD, metabolic dysfunction-associated steatotic liver disease; MetALD, metabolic and alcohol-associated liver disease.

**Table 4. Hazard ratios (HRs) for incident MASLD according to weekly working hours: results from time-dependent Cox regression.**

| Weekly working hours | Age- and sex-adjusted HR (95% CI) | Multivariable-adjusted HR (95% CI) | |
|---|---|---|---|
| | | Model 1* | Model 2** |
| ≤40 | 1.00 (reference) | 1.00 (reference) | 1.00 (reference) |
| 41–52 | 1.071 (1.046–1.096) | 1.066 (1.041–1.091) | 1.077 (1.050–1.106) |
| ≥53 | 1.393 (1.342–1.446) | 1.375 (1.325–1.428) | 1.378 (1.322–1.435) |

*Model 1: Adjusted for age, sex, smoking status, and regular exercise.

**Model 2: Model 1 plus adjustment for education level, marital status, household income, and shift work schedule.

CI, confidence interval; HR, hazard ratio; MASLD, metabolic dysfunction-associated steatotic liver disease.

## Discussion

Our findings suggest that even in individuals without prior liver disease or apparent predisposition, long working hours may contribute to the development of steatotic liver disease. Notably, the association persisted when restricted to MASLD alone but was less apparent for MASLD+MetALD. This discrepancy may be partly explained by the higher prevalence of heavy drinking among long-hour workers, which could have attenuated the observed association when alcohol consumption was considered. Specifically, the risk of developing MASLD was 1.17 times higher for participants working ≥60 hours per week than for those working 35–59 hours per week. When stratified by sex and age, long working hours were significantly associated with steatotic liver disease in men and in younger participants. However, after applying Bonferroni correction for multiple subgroup comparisons, the association persisted only among younger participants, suggesting that age may modify the effect of long working hours on MASLD risk.

Metabolic dysregulation, one of the main mechanisms underlying steatotic liver disease, is well-reflected in the MASLD classification [12]. Specifically, because other etiologies of liver disease such as viral infection and the use of steatogenic medications were excluded from the baseline, the cardiometabolic component of steatotic liver disease was more evident in this study. Among metabolic components, obesity [19], diabetes mellitus (DM) [20,21], hypertension [22], dyslipidemia [23,24], and metabolic syndrome [25] are factors that have been reported to be related to long working hours. Our results are consistent with those of previous studies and suggest that metabolic dysfunction plays a critical role in the association between long working hours and the development of steatotic liver disease. A previous cross-sectional study involving 22,818 Korean workers reported that long working hours were associated with MASLD in male workers [14]. However, as this study is based on data collected at a single point in time, determining a causal relationship between extended working hours and MASLD is difficult. Moreover, the use of the Hepatic Steatosis Index—an indirect and noninvasive method—may reduce diagnostic precision. To our knowledge, this is the first longitudinal investigation to clarify the time-based association between prolonged work hours and the development of MASLD using imaging-based criteria.

Several biological mechanisms underlying the association between long working hours and incident MASLD should be considered. Long working hours induce psychological stress [26]; chronic stress disrupts the hypothalamic–pituitary–adrenal axis, resulting in prolonged cortisol secretion [27], which, in turn, increases insulin resistance [28]. In turn, insulin resistance promotes hepatic lipogenesis [29]. In addition, chronic stress triggers the release of pro-inflammatory cytokines, including interleukin-6 and tumor necrosis factor-alpha [30], which increase insulin resistance and cause chronic low-grade inflammation, contributing to oxidative stress and subsequent liver injury [31]. Recently, the multiple-hit hypothesis has been proposed, and one of its components is disruption of the gut–liver axis [31,32]. This pathway remains hypothetical in our study, as we did not directly measure gut microbiota or intestinal permeability. Nevertheless, it provides a plausible framework for understanding how long working hours might contribute to systemic inflammation and hepatic injury. Regarding behavioral mechanisms, sleep deprivation resulting from long working hours induces insulin resistance [33] and increases appetite by reducing leptin and elevating ghrelin levels, subsequently promoting weight gain [34]. In addition,

sleep deprivation interferes with the circadian regulation of hepatic clock genes, leading to impaired metabolic homeostasis, which contributes to triglyceride accumulation and inflammation in the liver [35]. Irregular eating habits can also lead to circadian misalignments, thereby increasing the risk of MASLD [36]. Finally, prolonged sedentary behavior and insufficient physical activity may lead to the development of MASLD mediated by metabolic syndrome [37,38]. Beyond these biological and behavioral mechanisms, the MASLD classification itself provides additional clinical significance. By requiring both hepatic steatosis and at least one cardiometabolic risk factor, MASLD offers a more clinically relevant framework than NAFLD. This definition enables earlier identification of individuals at increased risk for cardiovascular and metabolic diseases, improves risk stratification, and facilitates preventive interventions, thereby aligning liver disease management with broader strategies in cardiometabolic medicine and public health. Men may be more vulnerable to MASLD associated with long working hours, for several reasons. Estrogen has a protective effect against hepatic fat accumulation, as illustrated by the increased severity of steatotic liver disease in postmenopausal women [39,40]. Furthermore, compared to women, men tend to have a higher proportion of visceral fat relative to their body mass index (BMI), predisposing them to a higher risk of steatotic liver disease [40]. From a behavioral perspective, men are more likely than women to engage in unhealthy behaviors, such as smoking, alcohol consumption, and unhealthy diets, which contribute to the development of steatotic liver disease [41]. The risk of steatotic liver disease may also have been underestimated because unpaid labor, such as childcare and housework, was not reflected in the reported working hours. However, the mechanism underlying the increased risk of MASLD in relatively young workers remains unclear. Previous studies have found that sleep deprivation in young individuals has been associated with lower leptin and higher ghrelin levels, and that they are more vulnerable to the adverse effects of sleep deficiency and circadian misalignment [34,42]. Additionally, the healthy worker effect should be taken into account, as older individuals who remain in the workforce are generally healthier and more health-conscious than their non-working counterparts, which may lead to an underestimation of the true risk of steatotic liver disease. Consequently, further research is needed to refine the study design and elucidate the underlying mechanisms.

Our study has several limitations. First, as information on work-related and lifestyle variables was collected through self-administered questionnaires, the possibility of measurement error cannot be ruled out. However, because the participants had no incentive to underreport or overreport for personal gain, the impact of differential misclassification on the results was likely to be minimal. Second, steatotic liver disease was identified through abdominal ultrasonography instead of definitive diagnostic procedures such as liver biopsy. However, liver biopsy, while considered the diagnostic gold standard, is invasive and carries the risk of complications. In large-scale epidemiological research, abdominal ultrasonography is commonly used due to its non-invasive nature, practicality, and acceptable accuracy in detecting steatotic liver disease [43]. Third, there were limitations in occupational variables, because occupations could not be categorized into groups such as white-collar and blue-collar occupations. In addition, the recent rise in telecommuting and flexible working arrangements may not have been fully captured by self-reported weekly working hours. These issues should be addressed in future studies. Fourth, selection bias among the final study population should be considered, because a greater number of individuals with missing data on working hours or ultrasonography were excluded from the study. However, the baseline characteristics did not differ significantly between the included and excluded groups, suggesting that the impact of this potential bias was likely minimal. Finally, our study population consisted of young and middle-aged Korean workers with a relatively good health status and high educational attainment. This demographic profile may limit the generalizability of our findings. Therefore, caution should be exercised when applying these results to populations with different age distributions, lower socioeconomic status, or varying racial and ethnic backgrounds. In addition, our dataset did not capture female-specific factors such as unpaid domestic labor, reproductive status, or hormonal changes, which may influence susceptibility to steatotic liver disease. As a result, we could not fully explore mechanisms underlying sex differences, and future studies incorporating such variables will be needed to clarify the role of female-specific determinants.

Despite certain limitations, the present study offers several notable strengths. As far as we are aware, this is the first prospective investigation to reveal a time-dependent relationship between prolonged work hours and MASLD onset,

supported by a large sample, a well-established cohort framework, consistent data collection, and a decade-long follow-up. Additionally, the study focused on a relatively young and middle-aged healthy population, which may have helped minimize survival bias associated with chronic conditions. Lastly, by incorporating repeated assessments of working hours to define a stable exposure group, the study likely captured the effects of extended work hours more accurately than prior research. To address limitations of self-reported data and enhance the reliability of the risk estimation, we included only individuals whose working-hours categories remained unchanged throughout the follow-up period, thereby ensuring a more robust survival analysis.

Furthermore, we conducted a sensitivity analysis using a time-dependent Cox regression that included individuals whose working-hours categories changed over time. The association between long working hours and MASLD remained robust in this analysis, further supporting the reliability of our findings.

In conclusion, our large-scale cohort study demonstrated that long working hours are associated with an increased risk of incident steatotic liver disease, particularly MASLD. This association was especially evident among male and relatively young workers, suggesting that long working hours are an independent risk factor for MASLD. These findings also carry policy relevance in the context of Korea's working-hours reforms, underscoring that exceeding thresholds such as 52 or 60 hours per week may pose significant health risks.

## Supporting information

**S1 Table. Temporal distribution of participant exclusions at each stage of study population selection.**
(DOCX)

**S2 Table. Subgroup analyses of the association between long working hours and the risk of MASLD with Bonferroni-adjusted p-values.**
(DOCX)

## Acknowledgments

This study was conducted using data provided by the Kangbuk Samsung Health Study. The authors thank all study participants and personnel for their dedication and continuing support.

## Author contributions

**Conceptualization:** Woncheol Lee.

**Formal analysis:** Yesung Lee, Woncheol Lee.

**Investigation:** Yesung Lee, Woncheol Lee.

**Methodology:** Yesung Lee, Woncheol Lee.

**Writing – original draft:** Yesung Lee.

**Writing – review & editing:** Yesung Lee, Woncheol Lee.

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
