## [Decision Letter · Decision Letter 0]

10 Sep 2025

Dear Dr. Lee,

We look forward to receiving your revised manuscript.

Kind regards,

Anna Di Sessa, PhD, MD

Academic Editor

PLOS ONE

**Journal Requirements:**

1. When submitting your revision, we need you to address these additional requirements. Please ensure that your manuscript meets PLOS ONE's style requirements, including those for file naming. The PLOS ONE style templates can be found at https://journals.plos.org/plosone/s/file?id=wjVg/PLOSOne_formatting_sample_main_body.pdf and https://journals.plos.org/plosone/s/file?id=ba62/PLOSOne_formatting_sample_title_authors_affiliations.pdf 2. Thank you for stating in your Funding Statement: This work was supported by the National Research Foundation of Korea (NRF) grant funded by the Korea government (MSIT) (RS-2023-00274176). The funders had no role in study design, data collection and analysis, decision to publish, or preparation of the manuscript.   Please provide an amended statement that declares *all* the funding or sources of support (whether external or internal to your organization) received during this study, as detailed online in our guide for authors at http://journals.plos.org/plosone/s/submit-now.  Please also include the statement “There was no additional external funding received for this study.” in your updated Funding Statement. Please include your amended Funding Statement within your cover letter. We will change the online submission form on your behalf. 3. We note that you have indicated that there are restrictions to data sharing for this study. For studies involving human research participant data or other sensitive data, we encourage authors to share de-identified or anonymized data. However, when data cannot be publicly shared for ethical reasons, we allow authors to make their data sets available upon request. For information on unacceptable data access restrictions, please see http://journals.plos.org/plosone/s/data-availability#loc-unacceptable-data-access-restrictions.  Before we proceed with your manuscript, please address the following prompts: a) If there are ethical or legal restrictions on sharing a de-identified data set, please explain them in detail (e.g., data contain potentially identifying or sensitive patient information, data are owned by a third-party organization, etc.) and who has imposed them (e.g., a Research Ethics Committee or Institutional Review Board, etc.). Please also provide contact information for a data access committee, ethics committee, or other institutional body to which data requests may be sent. b) If there are no restrictions, please upload the minimal anonymized data set necessary to replicate your study findings to a stable, public repository and provide us with the relevant URLs, DOIs, or accession numbers. Please see http://www.bmj.com/content/340/bmj.c181.long for guidelines on how to de-identify and prepare clinical data for publication. For a list of recommended repositories, please see https://journals.plos.org/plosone/s/recommended-repositories. You also have the option of uploading the data as Supporting Information files, but we would recommend depositing data directly to a data repository if possible. Please update your Data Availability statement in the submission form accordingly. 4. When completing the data availability statement of the submission form, you indicated that you will make your data available on acceptance. We strongly recommend all authors decide on a data sharing plan before acceptance, as the process can be lengthy and hold up publication timelines. Please note that, though access restrictions are acceptable now, your entire data will need to be made freely accessible if your manuscript is accepted for publication. This policy applies to all data except where public deposition would breach compliance with the protocol approved by your research ethics board. If you are unable to adhere to our open data policy, please kindly revise your statement to explain your reasoning and we will seek the editor's input on an exemption. Please be assured that, once you have provided your new statement, the assessment of your exemption will not hold up the peer review process. 5. Your ethics statement should only appear in the Methods section of your manuscript. If your ethics statement is written in any section besides the Methods, please move it to the Methods section and delete it from any other section. Please ensure that your ethics statement is included in your manuscript, as the ethics statement entered into the online submission form will not be published alongside your manuscript. 6. We note that you have referenced (Kivimäki M, Virtanen M, Kawachi I, Nyberg ST, Alfredsson L, Batty GD, et al.) which has currently not yet been accepted for publication. Please remove this from your References and amend this to state in the body of your manuscript: (Kivimäki M, Virtanen M, Kawachi I, Nyberg ST, Alfredsson L, Batty GD, et al. [Unpublished]) as detailed online in our guide for authorshttp://journals.plos.org/plosone/s/submission-guidelines#loc-reference-style 7. If the reviewer comments include a recommendation to cite specific previously published works, please review and evaluate these publications to determine whether they are relevant and should be cited. There is no requirement to cite these works unless the editor has indicated otherwise. 

**Additional Editor Comments:**

Although the paper has potential, it would benefit from major revisions before it can be considered for publication. All the issues raised by the reviewers need to be carefully addressed.

Reviewers' comments:

Reviewer's Responses to Questions

**Comments to the Author**

1. Is the manuscript technically sound, and do the data support the conclusions?

Reviewer #1: Yes

Reviewer #2: Yes

Reviewer #3: Yes

2. Has the statistical analysis been performed appropriately and rigorously?

Reviewer #1: Yes

Reviewer #2: Yes

Reviewer #3: Yes

3. Have the authors made all data underlying the findings in their manuscript fully available?

Reviewer #1: Yes

Reviewer #2: Yes

Reviewer #3: Yes

4. Is the manuscript presented in an intelligible fashion and written in standard English?

Reviewer #1: Yes

Reviewer #2: Yes

Reviewer #3: Yes

**Reviewer #1:**  The article represents a pioneering effort in applying the MASLD criteria to a large-scale cohort, thereby addressing a gap identified in prior research, such as the NAFLD study by Lee et al. (2021). While the core data presented in the paper are reliable, several aspects require standardization to enhance the academic rigor of the presentation.

1. The article does not sufficiently establish the origins of the term "metabolic dysfunction-associated steatohepatopathy" (MASLD) in the Introduction and Methods sections. It is crucial to reference Rinella et al. (2023), which is the inaugural international Delphi Consensus that established the terminology and diagnostic criteria for MASLD, replacing the former NAFLD classification. Properly tracing the terminology to its origin is recommended to prevent subjective definitions of diagnostic criteria.

2.The article's mechanism partially relies on previous studies without introducing novel pathways. It is advised to underscore the clinical significance of the MASLD classification in the Discussion section.

3.The paper lacks a prospective efficacy analysis, and it is recommended to supplement this with a detailed basis for calculation. The Cox model was validated but lacked multiple comparison corrections; a Bonferroni correction is advised.

3.The flowchart should label the temporal distribution of excluded cases to check for selection bias. Table headings were misplaced (P5 Table1 should be above the table); reference 17 (Ryu et al. 2015) was not cited in the text; and the abbreviation MASLD was not initially defined. A thorough review is recommended for the next manuscript to prevent these errors.

**Reviewer #2: ** Dear authors, Thank you for this interesting manuscript. here are my comments for more clarity:

1. Introduction and Background

The introduction effectively sets the context with global and Korean-specific data on working hours. However, it could briefly elaborate on why MASLD is a priority outcome (e.g., its rising prevalence and links to cardiometabolic disease beyond NAFLD). Please add 1–2 sentences in lines 58–66 referencing the global burden of MASLD (e.g., cite a recent review on its epidemiology) to strengthen the rationale.

2. Methods

o In lines 118–128, please specify if overtime was explicitly included in the self-reported question or if it relies on participant interpretation.

o For subgroup analyses (lines 174–177), it is better to justify the age cutoff (≤47 vs. ≥48 years) more explicitly—e.g., if it is based on the 90th percentile to capture younger vs. older workers.

o Please add a sentence on power calculations or sample size justification, given the low number of long-hour workers (n=2,243 vs. 115,111 in the reference group).

3. Results

o Please ensure consistency in reporting: The abstract says "higher risk of developing MASLD" (HR 1.17), but specify it's for ≥60 vs. 35–59 hours.

4. Discussion

o It is better to discuss policy implications more explicitly: Given Korea's work-hour reforms (Ref. 4), how might these findings inform thresholds like 52 or 60 hours/week?

o Please add a brief note on why the association persisted in MASLD alone vs. MASLD+MetALD, perhaps tying it to alcohol's role in long-hour workers (higher heavy drinking at baseline).

• Clarity and Writing: The manuscript is well-written overall

5. Generally

• Clarity and Writing: The manuscript is well-written overall, but some sentences are lengthy (e.g., lines 80–88). Break them for readability.

• Dates: IRB approval and data access are dated 2025 (lines 93–97), while data collection ends in 2023. If this is a projection or typo, clarify; otherwise, there is no issue given the query date (Sep 2025).

**Reviewer #3:**  I believe that the authors have exposed enormous and highly appreciated efforts in conducting this longitudinal cohort study to examine the association between long work hours and steatotic liver disease. However, some remarks should be considered:

THE TITLE:

the cohort study was conducted on patients being diagnosed with MASLD (with no alcohol or light alcohol intake), or MetALD (metabolic and alcohol-157 associated liver disease). both terms must be emphasized in the title and using “metabolic dysfunction-associated steatotic liver disease” alone will be misleading, since it means MASLD only. I suggest the following title “Effects of long working hours on Steatotic Liver Disease (MASLD and MetALD) in healthy workers: A 10-year cohort study”

ABSTRACT:

Since MetALD was being examined in this decade cohort study, the abstract did not reveal any information regarding those examined patients, regarding the objectives, results and conclusion sentences. I believe the authors should re-evaluate their manuscript in each details to be comprehensively summarized in the abstract.

INTRODUCTION:

MetALD disease should not be mentioned as part of MASLD system, and must be extensively differentiated in between, through providing etiologic manifestations of each disease (cause, criteria, alcohol intake), and main levels of their pathophysiologic progression. As well, recently published articles correlating the MetALD with workload or long working hours will be highly recommended to cite.

However, the objectives of the study must reflect the conducted work in this cohort study, and NOT depending on correlation of MASLD with work load only.

MATERIALS AND METHODS:

The study analyzed data from 117,354 participants, enhancing statistical power and generalizability within the Korean working population, and through applying rigorous exclusion criteria (i.e., excluding participants with confounding conditions (e.g., viral hepatitis, steatogenic medications)) has improved the internal validity. As well, depending on MASLD diagnosis via abdominal ultrasonography is more accurate than indirect indices like the Hepatic Steatosis Index.

• Sensitivity Analysis was performed in the study, through including models that account for time-varying exposure to test robustness. However, the authors should clarify if any of multi-pronged strategies being applied to enhance data accuracy and validity, such as workplace records (to obtain actual working hours from employer records), digital tools (e.g., swipe cards, login/logout systems to track work duration more accurately), Cross-Validation, or Repeated Measures (Collect working hours data at multiple time points to identify inconsistencies or trends).

• Moreover, limited occupational detail (e.g., white-collar vs. blue-collar), which could influence exposure and outcomes, exclusion of changing work hours during follow-up in the main analysis, potentially omitting relevant variability. In addition, behavioral factors like sleep and diet were discussed but not measured, limiting mechanistic insights.

RESULTS:

Using Cox proportional hazards models with multiple adjustments (age, sex, lifestyle, socioeconomic factors) have strengthened the findings, and subgroup analyses, through stratification by age and sex have provided nuanced insights into vulnerable populations in the detailed tables

• The authors should further discussed any possible residual confounding lead to the modest association of MASLD with long work hours (adjusted HR for MASLD was 1.17), and the argument regarding the bias being developed from the exclusion of participants with missing data was inadequate and need in-depth justification.

• While the sensitivity analysis used three categories of working hours, the main analysis used only two, limiting granularity. Besides, the models did not test for interaction effects (e.g., between sex and working hours), which could reveal synergistic risks.

DISCUSSION:

The authors provide a comprehensive explanation of physiological pathways linking long working hours to MASLD, and findings are contextualized with previous studies, including cross-sectional and meta-analytic evidence.

• Efforts must be paid to justify the absence of deep analysis of female-specific factors (e.g., unpaid labor, hormonal changes), while male vulnerability is well-discussed. Some proposed biological mechanisms (e.g., gut-liver axis disruption) are hypothetical and not directly measured, and further details regarding any unpublished analysis should be explained. In addition, the discussion lacks concrete suggestions for workplace or policy interventions.

**Do you want your identity to be public for this peer review?** For information about this choice, including consent withdrawal, please see our Privacy Policy

Reviewer #1: No

Reviewer #2: No

Reviewer #3: No

---

## [Author Response · Author response to Decision Letter 1]

30 Sep 2025

Response to the Academic Editor

1. Journal Style Requirements

Editor’s comment: When submitting your revision, we need you to address these additional requirements.

Response: We have revised the manuscript to comply with the PLOS ONE formatting guidelines. Headings and structure have been modified according to the required style.

2. Funding Statement

Editor’s comment: Please provide an amended statement that declares all sources of support, and include the sentence “There was no additional external funding received for this study.”

Response: We have revised the Funding Statement as requested. The sentence “There was no additional external funding received for this study.” has been included in the cover letter and the Funding Statement has been updated accordingly.

3. Data Sharing

Editor’s comment: Please clarify the reasons for data restrictions and provide details regarding conditions for access.

Response: We have updated the Data Availability statement to clearly explain the reasons why the data cannot be publicly shared and to specify the conditions under which de-identified data may be accessed. The revised statement has also been included in the Acknowledgments section of the manuscript.

4. Ethics Statement

Editor’s comment: The ethics statement should appear only in the Methods section.

Response: The ethics statement is now included exclusively in the Methods section, and duplicate statements in other sections have been removed.

5. Reference Marked as “Unpublished”

Editor’s comment: The reference (Kivimäki et al.) has currently not yet been accepted for publication. Please remove it from the References and cite it in the text as “Unpublished.”

Response: We would like to clarify that according to PubMed, the following article is indexed with publication details:

Kivimäki M, Virtanen M, Kawachi I, Nyberg ST, Alfredsson L, Batty GD, et al. Long working hours, socioeconomic status, and the risk of incident type 2 diabetes: a meta-analysis of published and unpublished data from 222,120 individuals. Lancet Diabetes Endocrinol. 2015 Jan;3(1):27–34. Epub 2014 Sep 25. doi: 10.1016/S2213-8587(14)70178-0.

Since this article appears in PubMed with a DOI and full citation, we would appreciate the Editor’s guidance on why it should be considered “unpublished.”

Response to Reviewer #1

We thank the reviewer for the positive assessment of our manuscript and for the constructive suggestions that helped us improve clarity, rigor, and consistency. We address each point below and indicate where changes have been made in the revised manuscript.

Comment 1

The article does not sufficiently establish the origins of the term "metabolic dysfunction-associated steatohepatopathy (MASLD)" in the Introduction and Methods. It is crucial to reference Rinella et al. (2023), which is the inaugural international Delphi Consensus that established the terminology and diagnostic criteria for MASLD, replacing the former NAFLD classification.

Response:

We thank the reviewer for this valuable suggestion. We have revised the Introduction to clearly trace the origins of the MASLD terminology and diagnostic framework to the 2023 international multisociety Delphi consensus (Rinella et al., 2023). We now emphasize that the consensus replaced the former NAFLD classification and established MASLD as the preferred term to highlight the cardiometabolic risk factors associated with steatotic liver disease. The text has been revised accordingly (Introduction, lines 36–44).

Comment 2

The mechanism section relies on prior studies without introducing novel pathways; underscore the clinical significance of the MASLD classification in the Discussion section.

Response:

We agree with the reviewer’s comment. To address this, we have added a new paragraph in the Discussion emphasizing the clinical significance of the MASLD classification. Specifically, we note that by requiring both hepatic steatosis and at least one cardiometabolic risk factor, MASLD provides a more clinically relevant framework than NAFLD. This definition allows for earlier identification of individuals at risk for cardiovascular and metabolic diseases, improves risk stratification, and facilitates preventive interventions. It also aligns liver disease management with broader strategies in cardiometabolic medicine and public health. These revisions can be found in the Discussion (lines 291-296).

Comment 3

The paper lacks a prospective efficacy analysis, and the Cox model was validated but lacked multiple comparison corrections; a Bonferroni correction is advised.

Response:

We appreciate the reviewer’s concern. In response to the reviewer’s recommendation, we applied Bonferroni correction to the subgroup analyses (sex and age strata, n=4). The association between long working hours and incident MASLD in participants aged ≤47 years remained statistically significant after correction (HR 1.20, 95% CI 1.05–1.38; adjusted p=0.031), whereas other subgroup analyses did not remain significant. These results have been added as Supplementary Table S2, and the corresponding text has been updated in the Methods, Results and Discussion sections (lines 170-171, 223-226, 254-257).

Comment 4

The flowchart should label the temporal distribution of excluded cases to check for selection bias. Table headings were misplaced (P5 Table1 should be above the table); reference 17 (Ryu et al. 2015) was not cited in the text; and the abbreviation MASLD was not initially defined. A thorough review is recommended for the next manuscript to prevent these errors.

Response:

We appreciate the reviewer’s careful assessment and have addressed each point as follows.

1. Flowchart (temporal distribution of exclusions).

Although temporal labeling is not routinely requested in flowcharts, we agree that providing this information improves transparency regarding potential selection bias. We therefore revised Figure 1 to indicate in the figure legend that exclusions occurred across the study period without clustering, and we added a new Supplementary Table S1 reporting the distribution of excluded participants by exclusion category across 3-year baseline periods. These additions allow readers to evaluate temporal patterns of exclusion. (Figure 1, legend; Supplementary Table S1).

2. Table headings.

We appreciate the reviewer’s careful comments. We have double-checked the formatting of Table 1 in the submitted manuscript and confirmed that the table heading is appropriately placed above the table, in accordance with journal guidelines.

3. Reference 17 (Ryu et al., 2015).

We thank the reviewer for the careful reading. Reference 17 (Ryu et al., 2015) is cited in the text (line 140). We have double-checked and confirmed that the citation is properly inserted.

4. Abbreviation.

The abbreviation MASLD is now defined at first mention in both the Abstract and Introduction, and is used consistently thereafter throughout the manuscript.

Response to Reviewer #2

1. Introduction and Background

Comment: The introduction effectively sets the context with global and Korean-specific data on working hours. However, it could briefly elaborate on why MASLD is a priority outcome (e.g., its rising prevalence and links to cardiometabolic disease beyond NAFLD). Please add 1–2 sentences in lines 58–66 referencing the global burden of MASLD (e.g., cite a recent review on its epidemiology) to strengthen the rationale.

Response:

We thank the reviewer for this helpful suggestion. We have revised the Introduction (lines 45-48) to highlight why MASLD is a priority outcome by emphasizing its global burden, rising prevalence, and clinical significance beyond NAFLD. In particular, we added that MASLD affects nearly 38% of adults worldwide and is projected to exceed 55% by 2040, with cardiovascular disease being the leading cause of death in affected individuals. To strengthen the rationale, we have cited recent epidemiological reviews (Younossi et al., 2025).

2. Methods

Comment 1: In lines 118–128, please specify if overtime was explicitly included in the self-reported question or if it relies on participant interpretation.

Response:

We appreciate the reviewer’s comment. We clarify that overtime was explicitly included in the self-reported question on weekly working hours. The Methods section (lines 108-109) now reads: “Working hours were assessed using the following question: ‘How many hours did you work in a week on average in your job for the past year, including overtime?’”

Comment 2: For subgroup analyses (lines 174–177), it is better to justify the age cutoff (≤47 vs. ≥48 years) more explicitly—e.g., if it is based on the 90th percentile to capture younger vs. older workers.

Response:

We agree and have revised the Methods (lines 121-123) to clarify that the cutoff corresponded to the 90th percentile of age in the study population, allowing us to distinguish younger from older workers.

Comment 3: Please add a sentence on power calculations or sample size justification, given the low number of long-hour workers (n=2,243 vs. 115,111 in the reference group).

Response:

We thank the reviewer for this suggestion. Although the number of long-hour workers was relatively small compared to the reference group, the overall cohort size ensured adequate statistical power for subgroup analyses. We added a sentence in the Methods (lines 171-173) to clarify this.

3. Results

Comment: Please ensure consistency in reporting: The abstract says "higher risk of developing MASLD" (HR 1.17), but specify it's for ≥60 vs. 35–59 hours.

Response:

We agree and have revised the Abstract (line 11-13) to specify that the comparison was between participants working ≥60 hours per week and those working 35–59 hours per week.

4. Discussion

Comment 1: It is better to discuss policy implications more explicitly: Given Korea's work-hour reforms (Ref. 4), how might these findings inform thresholds like 52 or 60 hours/week?

Response:

We appreciate this valuable suggestion. We have expanded the Discussion (lines 352-357) to explicitly address the policy implications of our findings in light of Korea’s work-hour reforms, noting that our results support the importance of monitoring thresholds such as 52 and 60 hours per week.

Comment 2: Please add a brief note on why the association persisted in MASLD alone vs. MASLD+MetALD, perhaps tying it to alcohol's role in long-hour workers (higher heavy drinking at baseline).

Response:

We thank the reviewer for this insightful comment. We have added a brief note in the Discussion (lines 248-252) suggesting that differential alcohol consumption patterns among long-hour workers may partly explain the stronger association observed with MASLD alone compared with MASLD+MetALD.

5. Generally

Comment 1: Clarity and Writing: The manuscript is well-written overall, but some sentences are lengthy (e.g., lines 80–88). Break them for readability.

Response:

We agree with the reviewer’s comment. The long sentence in lines 80–88 has been revised and divided into shorter sentences to improve clarity and readability.

Comment 2: Dates: IRB approval and data access are dated 2025 (lines 93–97), while data collection ends in 2023. If this is a projection or typo, clarify; otherwise, there is no issue given the query date (Sep 2025).

Response:

We appreciate the reviewer’s careful attention. The Kangbuk Samsung Health Study is an ongoing cohort with continuous data collection. The Cohort Data Center periodically organizes the accumulated data and releases datasets to authorized researchers. For this study, data collected up to 2023 were available, while 2024 data are currently under processing. We clarified this in the Methods section (lines 65-67), noting that IRB approval for the present analysis was obtained in 2025 after data linkage was completed.

Response to Reviewer #3

Comment 1:

THE TITLE: the cohort study was conducted on patients being diagnosed with MASLD (with no alcohol or light alcohol intake), or MetALD (metabolic and alcohol-157 associated liver disease). both terms must be emphasized in the title and using “metabolic dysfunction-associated steatotic liver disease” alone will be misleading, since it means MASLD only. I suggest the following title “Effects of long working hours on Steatotic Liver Disease (MASLD and MetALD) in healthy workers: A 10-year cohort study”

Response:

We thank the reviewer for the helpful suggestion regarding the title. To more accurately reflect the study outcomes, we have revised the title to: “Effects of long working hours on metabolic dysfunction-associated steatotic liver disease, with and without increased alcohol intake, in healthy workers: A 10-year cohort study.”

Comment 2:

ABSTRACT: Since MetALD was being examined in this decade cohort study, the abstract did not reveal any information regarding those examined patients, regarding the objectives, results and conclusion sentences. I believe the authors should re-evaluate their manuscript in each details to be comprehensively summarized in the abstract.

Response:

We thank the reviewer for this valuable comment. We agree that the Abstract should comprehensively reflect the outcomes examined in our study. We have revised the Abstract accordingly. Specifically, the Objectives now state that the study assessed the association of long working hours with steatotic liver disease including both MASLD and MASLD+MetALD. In the Results, we added the hazard ratios for MASLD+MetALD alongside those for MASLD. Finally, in the Conclusion, we clarified that prolonged working hours were associated with an increased risk of steatotic liver disease, including both MASLD and MASLD+MetALD, particularly among younger and male participants.

Comment 3:

INTRODUCTION: MetALD disease should not be mentioned as part of MASLD system, and must be extensively differentiated in between, through providing etiologic manifestations of each disease (cause, criteria, alcohol intake), and main levels of their pathophysiologic progression. As well, recently published articles correlating the MetALD with workload or long working hours will be highly recommended to cite. However, the objectives of the study must reflect the conducted work in this cohort study, and NOT depending on correlation of MASLD with work load only.

Response:

We thank the reviewer for this important comment. We agree that MASLD and MetALD should be clearly distinguished, as they represent distinct disease categories within the recent consensus framework. Accordingly, we revised the Introduction to better differentiate MASLD and MetALD. We also clarified in the Objective statement that our study investigated the association between long working hours and steatotic liver disease, including both MASLD and MASLD+MetALD, rather than MASLD alone.

Comment 4:

MATERIALS AND METHODS:

The study analyzed data from 117,354 participants, enhancing statistical power and generalizability within the Korean working population, and through applying rigorous exclusion criteria (i.e., excluding participants with confounding conditions (e.g., viral hepatitis, steatogenic medications)) has improved the internal validity. As well, depending on MASLD diagnosis via abdominal ultrasonography is more accurate than indirect indices like the Hepatic Steatosis Index.

• Sensitivity Analysis was performed in the study, through including models that account for time-varying exposure to test robustness. However, the authors should clarify if any of multi-pronged strategies being applied to enhance data accuracy and validity, such as workplace records (to obtain actual working hours from employer records), digital tools (e.g., swipe cards, login/logout systems to track work duration more accurately), Cross-Validation, or Repeated Measures (Collect working hours data at multiple time points to identify inconsistencies or trends).

• Moreover, limited occupational detail (e.g., white-collar vs. blue-collar), which could influence exposure and outcomes, exclusion of changing work hours during follow-up in the main analysis, potentially omitting relevant variability. In addition, behavioral factors like sleep and diet were discussed but not measured, limiting mechanistic insights.

Response:

We thank the reviewer for recognizing the metho

---

## [Decision Letter · Decision Letter 1]

29 Oct 2025

Effects of long working hours on metabolic dysfunction-associated steatotic liver disease, with and without increased alcohol intake, in healthy workers: A 10-year cohort study

PONE-D-25-36457R1

Dear Dr. Lee,

We’re pleased to inform you that your manuscript has been judged scientifically suitable for publication and will be formally accepted for publication once it meets all outstanding technical requirements.

Kind regards,

Anna Di Sessa, PhD, MD

Academic Editor

PLOS ONE

Additional Editor Comments (optional):

Reviewers' comments:

Reviewer's Responses to Questions

**Comments to the Author**

Reviewer #1: All comments have been addressed

Reviewer #3: All comments have been addressed

2. Is the manuscript technically sound, and do the data support the conclusions?

Reviewer #1: Yes

Reviewer #3: Yes

3. Has the statistical analysis been performed appropriately and rigorously?

Reviewer #1: Yes

Reviewer #3: Yes

4. Have the authors made all data underlying the findings in their manuscript fully available?

Reviewer #1: Yes

Reviewer #3: Yes

5. Is the manuscript presented in an intelligible fashion and written in standard English?

Reviewer #1: Yes

Reviewer #3: Yes

Reviewer #1: The authors have addressed all my comments. I have no other questions. The paper may be accepted in Plos One.

Thank you for your invitation.

Reviewer #3: I appreciate the authors’ detailed responses and the revisions made to address the reviewers’ concerns.

The authors have satisfactorily clarified the inquires regarding undetailed remarks in the abstract, introduction, methodology, results, discussion and conclusion sections through providing additional data, which strengthens the manuscript.

**Do you want your identity to be public for this peer review?** For information about this choice, including consent withdrawal, please see our Privacy Policy

Reviewer #1: **Yes: ** Zhigang Ren

Reviewer #3: No

---

## [Editor Report · Acceptance letter]

PONE-D-25-36457R1

PLOS ONE

Dear Dr. Lee,

I'm pleased to inform you that your manuscript has been deemed suitable for publication in PLOS ONE. Congratulations! Your manuscript is now being handed over to our production team.

Kind regards,

on behalf of

Dr. Anna Di Sessa

Academic Editor

PLOS ONE